

# Generalized Canonical Transform method for radio occultation sounding with improved retrieval in the presence of horizontal gradients

Michael Gorbunov[1,2], Gottfried Kirchengast[3], and Kent B. Lauritsen[4]

[1]A.M. Obukhov Institute of Atmospheric Physics, Russian Academy of Sciences, Pyzevsky per. 3, 119017, Moscow, Russia
[2]Hydrometcenter of Russia, Bol. Prechistensky per. 11-13, 123242, Moscow, Russia
[3]Wegener Center for Climate and Global Change (WEGC) and Institute for Geophysics, Astrophysics, and Meteorology/Institute of Physics, University of Graz, Brandhofgasse 5, 8010, Graz, Austria
[4]Danish Meteorological Institute, 2100, Copenhagen, Denmark

**Correspondence:** Michael Gorbunov (gorbunov@ifaran.ru)

**Abstract.** By now, a series of advanced Wave Optical (WO) approaches to the processing of Radio Occultation (RO) observations are widely used. In particular, the Canonical Transform (CT) method and its further developments need to be mentioned. The latter include the Full Spectrum Inversion (FSI) method, the Geometric Optical (GO) Phase Matching (PM) method, and the general approach based on the Fourier Integral Operators (FIOs), also referred to as the CT type 2 (CT2) method. The

5  general idea of these methods is the application of a canonical transform that changes the coordinates in the phase space from time and Doppler frequency to impact parameter and bending angle. For the spherically symmetric atmosphere, the impact parameter, being invariant for each ray, is a unique coordinate of the ray manifold. Therefore, the derivative of the phase of the wave field in the transformed space is directly linked to the bending angle, as a single-valued function of the impact parameter. However, in the presence of horizontal gradients, this approach may not work. Here we introduce a further generalization of

10  the CT methods in order to reduce the errors due to horizontal gradients. We describe, in particular, the modified CT2 method denoted CT2A, which complements the former with one more affine transform: a new coordinate that is a linear combination of the impact parameter and bending angle. The linear combination coefficient is a tunable parameter. We derive the explicit formulas for the CT2A and develop the updated numerical algorithm. For testing the method, we performed statistical analyses based on COSMIC RO retrievals and (collocated) ECMWF analysis profiles. We demonstrate that it is possible to find a rea-

15  sonably optimal value of the new tunable CT2A parameter that mitigates systematic errors in the lower troposphere and allows the practical realization of the improved capability to cope with horizontal gradients.



# 1 Introduction

The first step in the development of wave optical (WO) approach to the processing of radio occultation (RO) observations
was made by Melbourne et al. (1994), who used the thin screen approximation for the atmosphere combined with the Back
Propagation (BP) technique. This approach was further developed under the name of Fresnel Inversion by Mortensen and Høeg
(1998). Although the accuracy of this approximation in lower troposphere was insufficient for the practical application, its basic
idea was correct. It consisted in the reduction of the influence of the diffraction by using the BP, which made the inversion
results independent from the observation distance and canceled the resolution restriction due to the Fresnel zone size.

Later works (Gorbunov et al., 1996; Karayel and Hinson, 1997; Gorbunov and Gurvich, 1998a, b) developed a different
understanding of the BP technique. The BP wave field evaluated in some plane was not considered as the actual wave field, but
as a representation of the original field observed at the Low Earth orbit (LEO): in this representation, the effects of diffraction
and multipath propagation were significantly reduced. This, in a straightforward way, allowed evaluation of the geometric
optical (GO) bending angle profile, which was inverted in the framework of the standard GO scheme (Ware et al., 1996;
Kursinski et al., 1997).

The further development of the WO approach based on the representation view relied upon the concept of the Canonical
Transform (CT) originating from the classical mechanics (Arnold, 1978; Goldstein et al., 2014), generalized for the quantum
mechanics by Fock (1978), mathematically substantiated by Egorov (1985); Egorov and Shubin (1993). Further on this concept
obtained an extensive mathematical development (Treves, 1982a, b; Hörmander, 1985a, b). The correspondence between the
quantum and classical mechanics is the same as the link between the wave optics and geometrical optics.

In both cases, there is a strict mathematical representation (quantum mechanics or wave optics) and its asymptotic solution
(classical mechanics or geometrical optics). While the evolution of de Broglie waves of probability or electromagnetic waves is
described by the Hamilton operator, the evolution of rays or classical trajectories of particles is described by Hamilton system,
where the Hamilton operator is obtained by the substitute of the momentum operator instead of classical momentum. Accord-
ingly, for the classical problem the phase space is introduced, the dimension of which equals doubled geometric dimension,
because to each geometrical coordinate we can conjugate the corresponding momentum. For the wave problems momentum is
understood as the ray direction vector.

The canonical transforms arise, when we consider the class of the transforms of the phase space that conserve the canonical
form of the Hamilton dynamical system. It was first demonstrated by Fock (1978) that these transforms have a very simple
implementation in the quantum mechanics: they correspond to linear transforms of the wave function. The kernel of this
transform is derived in classical terms, but, still, it describes a short-wave asymptotical solution of the wave problem. This
idea was later mathematically developed first by Egorov (1985); Egorov and Shubin (1993) and then by Treves (1982a, b);
Hörmander (1985a, b).

The application of the CT approach for the RO observation processing was pioneered by Gorbunov (2002), where it was com-
bined with the BP. The idea of the CT without BP was first developed by Jensen et al. (2003, 2004) and later the general view at
these results in the framework of the CT approach was developed by Gorbunov and Lauritsen (2004a); Gorbunov et al. (2004).





Finally, it was recognized that the different methods: CT (Gorbunov, 2002), Full-Spectrum Inversion (FSI) (Jensen et al., 2003), Phase Matching (PM) (Jensen et al., 2004), and CT of the 2nd type (CT2) (Gorbunov and Lauritsen, 2004a) were, in fact, different approximations of the same solution, for which Fourier Integral Operators (FIOs) provided the general transform
approach (Gorbunov and Lauritsen, 2004a).

The idea of the CT approach is as follows. Given the observations or RO complex signal $u(t)$ as function of time $t$, which can be represented through its amplitude $A(t)$ and phase $\phi(t)$, $u(t) = A(t)\exp(i\phi(t))$. It is convenient to use eikonal, or phase path $\Psi(t) = \phi(t)/k$, where $k = 2\pi/\lambda$ is the wavenumber, and $\lambda$ is the wavelength. Thus, $u(t) = A(t)\exp(ik\Psi(t))$, and $k$ is the large parameter. The signal is composed of multiple sub-signals $u_i(t) = A_i(t)\exp(ik\Psi_i(t))$ corresponding to
interfering rays. For each sub-signal it is possible to introduce the instantaneous frequency $k\dot{\Psi}_i = k\sigma_i$. However, instantaneous frequency cannot be introduced for their composition.

The multipath propagation problem consists in the de-composition of the signal equal to the sum or different sub-signals, to retrieve the ray structure of the observed field. The solution of this problem discussed in the aforementioned papers consisted in the transform of the observed wave field $u(t)$ into a different representation. The new coordinates in the transformed space
were the ray impact parameter $p$ and bending angle $\epsilon$. The transform $(t, \sigma) \to (p, \epsilon)$ is canonical (Gorbunov and Lauritsen, 2004a), which allows for writing the corresponding linear transform $\widehat{\Phi}_2$, where the subscript 2 indicates that it is a CT of the 2nd type (Arnold, 1978; Goldstein et al., 2014), that maps the original field $u(t)$ to field in the impact parameter representation $\hat{u}(p) = \widehat{\Phi}_2[u(t)](p)$. The idea of the choice of the ray impact parameter as the new coordinate is based on the fact that in a spherically-symmetric medium, ray impact parameter is the ray invariant, which is known the Bouger law. The
locally spherically-symmetric medium is the basic approximation used in the inversion of RO data. For the real atmosphere with horizontal gradients, the dynamic equation for $p$ was derived by Gorbunov and Kornblueh (2001), who demonstrated that derivative of $p$ with respect to the ray arc length is equal to the horizontal component of the refractivity gradient in the occultation plane. Strong horizontal gradients may result in the situation when dependence $\epsilon(p)$ becomes multi-valued (Healy, 2001; Gorbunov and Lauritsen, 2009), which was referred to as the impact multipath (Zou et al., 2019).

The idea explored in the present manuscript consists in the further development of the CT approach by using a generalized transform with the coordinate $p' = p + \beta\epsilon$. Unlike the standard CT approach, where the form of the new coordinates in the phase is known in advance, this transform has the tunable parameter $\beta$ that can take into account the statistical impact multipath effect.

The paper is organized as follows. In Section 2 we discuss the canonical transform in wave optics and quantum mechanics in general terms, including brief review of FIOs. Based on this context we discuss in Section 3 the application of the CT method
for RO and introduce the particular phase space and the specific choice of coordinates as well as the new generalization adding an affine transform with a tunable parameter for improved the coping capability with horizontal gradients. In Section 4 we discuss the practical modifications needed to readily advance existing numerical implementations of the CT algorithm and present results of our performance evaluation from processing real-observed COSMIC RO data, including how to find an optimal value of the tunable parameter minimizing the systematic errors in the lower troposphere. Section 5 finally provides
the summary and main conclusions of the paper.





## 2 General concept of Canonical Transform in Wave Optics

We will start with a brief discussion of the Canonical Transform (CT). This concept originated from the classical mechanics (Arnold, 1978; Goldstein et al., 2014), where it referred to a kind of transform of the coordinates and momenta that conserve the Hamiltonian form of the dynamical equation. Fock (1978) introduced the CT in the quantum mechanics. Note, the first Russian edition of the monograph Fock (1978) appeared as early as in 1929. Because the relation between the classical and quantum mechanics, on one side, and the relation between the geometrical and wave optics, on the other side are the same, we can immediately apply the approach introduced by Fock (1978).

We assume that the wave field is can be represented in the standard form:

$$u(t) = A(t)\exp(ik\Psi(t)),$$

(1)

where $t$ is the observation time, $\Psi(t)$ is the eikonal, $k = 2\pi/\lambda$ is the wavenumber, $\lambda$ is the wavelength, $A(t)$ is the amplitude. The time $t$ can be associated with a specific spatial location of the observation, as it is the case in RO, but $u(t)$ can also be looked at as general signal.

The amplitude $A(t)$ and the derivative of $\Psi(t)$ are assumed to be slowly changing within an oscillation period. In this case, the wave field is termed quasi-monochromatic with an instant amplitude $A(t)$ and frequency $\omega(t) = k\dot{\Psi}(t)$. Otherwise, more generally, the field should be equal to a super-position of quasi-monochromatic components:

$$u(t) = \sum_j A^{(j)}(t)\exp\left(ik\Psi^{(j)}(t)\right),$$

(2)

where the upper index $j$ enumerates the components, $A^{(j)}(t)$ are their amplitudes, and $\Psi^{(j)}(t)$ are their eikonals. Each component has its own instant amplitude and frequency.

When discussing the CTs, it is necessary to bear in mind that most of the relations have an asymptotic nature, where $k$ is the large parameter (or $\lambda$ is the small parameter). The reason is as follows. Given measurements of wave field, each monochromatic component can be interpreted in terms of wave fronts and rays. Each point has a single ray, and its direction is linked to the normalized frequency $\sigma(t) = \dot{\Psi}(t)$. To this end, it is also necessary to know the position of the transmitter and receiver, as it takes place in RO observation. However, at this stage of the consideration of the problem, we can simply speak about instant tones of the signal.

Therefore, for a specific class of signals, including quasi-monochromatic ones and their superposition, it is possible to introduce a phase space $(t, \sigma)$. Although the original signal is 1-D, this space is 2-D, and the structure of the signal can be described in terms of the function $\sigma(t)$ which can be both single-valued for quasi-monochromatic signals, or multi-valued for superpositions of such signals.

Consider RO observations as an example. The original signal corresponds to a range of rays starting at the transmitter and the phase space $\sigma(t)$ is a very smooth continuous line. As the signal propagates through the atmosphere its structure gets more and more complicated. Still, in the phase space its topological structure remains the same: it is always a single continuous line, although it may have multiple projections to the axis of time $t$, which corresponds to multipath propaga-





tion (Gorbunov, 2002; Gorbunov and Lauritsen, 2004a). Such a line representing the signal structure is referred to as the ray manifold (Mishchenko et al., 1990).

The outstanding and, still, simple idea of Fock (1978) was that the classical canonical transforms correspond to linear integral transforms of the wave field with oscillating kernels. This class of transforms was later named Fourier Integral Operators (FIO) (Egorov, 1985; Egorov and Shubin, 1993; Treves, 1982a, b; Hörmander, 1985a, b). The general form of such an operator first discussed by Fock (1978) has the following form:

$$\hat{u}(p) = \sqrt{-\frac{ik}{2\pi}} \int a_2(p,t) \exp\left(ikS_2(p,t)\right) u(t) \, dt \equiv \Phi_2\left[u(t)\right](p), \tag{3}$$

where $p$ is a new coordinate in the mapped space. We use notation $\Phi_2$ and, accordingly, $a_2$ and $S_2$, because this type of operators was referred to as the FIO of the second type (Gorbunov and Lauritsen, 2004a), while the FIO of the first type is the composition of a Fourier transform and a second-type FIO (Egorov, 1985; Egorov and Shubin, 1993). This type of operators is linked to the corresponding type of the generating function (Arnold, 1978; Goldstein et al., 2014). Note, historically FIO of the second type appeared first, but in mathematical works it was FIO of the first type that were discussed first.

Considering now $u(t)$ as a quasi-monochromatic signal, we can derive the asymptotic form of transform (3) using the stationary phase principle:

$$\hat{u}(p) = \sqrt{-\frac{ik}{2\pi}} \int a_2(p,t) A(t) \exp\left(ik(S_2(p,t) + \Psi(t))\right) dt \equiv \Phi_2\left[u(t)\right](p), \tag{4}$$

The stationary phase point $t_s(p)$ of this integral satisfies the equation:

$$\frac{\partial}{\partial t} S_2(p,t) + \dot{\Psi}(t) = 0. \tag{5}$$

Accordingly, the transformed field, under the assumption that the Eq. (5) has a single solution $t_s(p)$, is also quasi-monochromatic and can be written as follows:

$$\hat{u}(p) = A'(p) \exp\left(ik\Psi'(p)\right) = A'(p) \exp\left(ik\left(S_2(p,t_s(p)) + \Psi(t_s(p))\right)\right). \tag{6}$$

Its instant frequency equals:

$$\xi(p) = \dot{\Psi}'(p) = \frac{d}{dp}\left(S_2(p,t_s(p)) + \Psi(t_s(p))\right) =$$

$$= \frac{\partial}{\partial p} S_2(p,t_s(p)) + \left(\frac{\partial}{\partial t} S_2(p,t_s(p)) + \frac{\partial}{\partial t} \Psi(t_s(p))\right) \frac{dt_s}{dp} =$$

$$= \frac{\partial}{\partial p} S_2(p,t_s(p)), \tag{7}$$

by virtue of Eq. (5). Recalling that $\dot{\Psi}(t) = \sigma$, which is the original momentum, we have the following relation between the canonical coordinates $(t,\sigma)$ and $(p,\xi)$, in the original and mapped spaces:

$$\frac{\partial S_2}{\partial t} = -\sigma, \quad \frac{\partial S_2}{\partial p} = \xi, \tag{8}$$





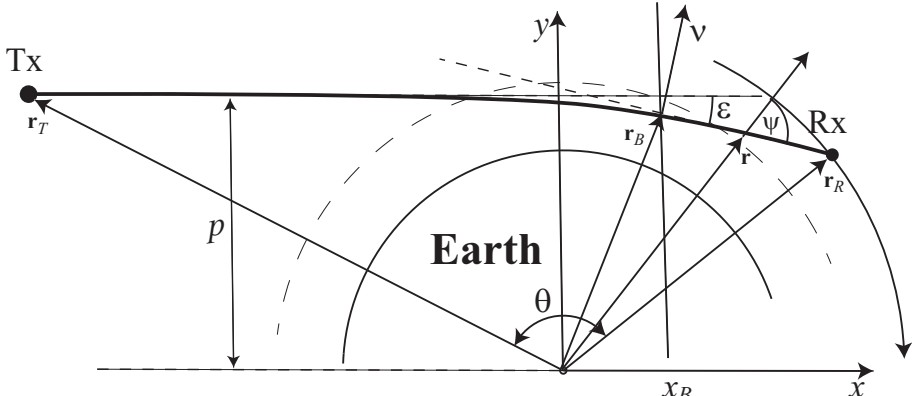

**Figure 1.** Radio occultation observation geometry with relevant geometrical variables indicated (for description see Sect. 3.1)

which can be expressed in terms of the differential $dS_2$:

$$dS_2 = \xi dp - \sigma dt. \tag{9}$$

And, vice versa, the requirement that the right-hand part in Eq. (9) should be equal to a full differential $dS_2$ of a function $S_2(p,t)$ is a necessary and sufficient condition for the transform $(t,\sigma) \rightarrow (p,\sigma)$ to be canonical (Arnold, 1978; Goldstein et al., 2014). The function $S_2(p,t)$ is then termed the generating function of the canonical transform.

In terms of FIO, $S_2(p,t)$ is referred to as its phase function, and $a_2(p,t)$ is its amplitude function. The phase function, which specifies the canonical transform, is of primary importance, while the amplitude function is derived using the energy conservation (Gorbunov and Lauritsen, 2004a). We see, therefore, that using the classical, or geometric optical concepts, it is possible to write down the asymptotic form of the quantum, or wave optical operator implementing the transformation of the original signal into a different representation. If the structure of the original signal is represented as a ray manifold in the phase

plane, such a transform is applied to the coordinates in this space. In particular, it may be possible to find such a coordinate system, where the ray manifold geometry will be exceptionally simple.

## 3    The Canonical Transform method for RO and its generalization

Here we discuss the application of the CT technique for the analysis of RO observations (Fig. 1) by first reviewing the different existing variants (3.1) and then introducing the new generalized CT method (3.2) and an application-relevant formulation for

readily updating existing algorithms (3.3).

### 3.1    Canonical Transform method in different existing variants

The RO observation geometry is schematically represented in Figure 1. The wave emitted by a transmitter Tx is received by a receiver Rx on a low-Earth orbit. Transmitter is borne by a satellite belonging to one of the modern Global Navigation





Satellites Systems (GNSS), including GPS, GLONASS etc. Due to the movement of the transmitter and receiver the ray
descends or ascends in the atmosphere, which allows the derivation of the atmospheric profiles from the bending angles $\epsilon(p)$
(Ware et al., 1996; Kursinski et al., 1997). The CT technique is used for the retrieval of bending angle profile from the wave
field measurements.

The first approach of processing RO data, belonging to the class of CT, was the Back Propagation (BP) (Gorbunov et al.,
1996; Karayel and Hinson, 1997; Gorbunov and Gurvich, 1998a, b). In this technique the field was linearly transformed to be
re-calculated to the BP plane locate at coordinate $x_B$:

$$u_B(y) = \sqrt{\frac{ik}{2\pi}} \int \frac{u(t)\exp(-ik|\mathbf{r}_B(y)-\mathbf{r}_R(t)|)}{|\mathbf{r}_B(y)-\mathbf{r}_R(t)|^{1/2}} |\sin\phi(\dot{\mathbf{r}}_R(t),\mathbf{r}_B(y)-\mathbf{r}_R(t))\,\dot{\mathbf{r}}_R(t)|\,dt, \qquad (10)$$

where 2-D vector $\mathbf{r}_B(y)$ equals $(x_B,y)$, $\phi(\mathbf{a},\mathbf{b})$ is the angle between vectors $\mathbf{a}$ and $\mathbf{b}$. This transform is performed under the
application of the procedure of the stationarization of the transmitting satellite and projection of the satellite movement to the
vertical plane. Note, the same procedure is commonly applied when using CT-like approaches. It is important that the BP field
is not the real field in the BP plane, because the BP procedure assumes the vacuum propagation. This procedure results in
some representation of the original wave field with reduced diffraction effects due to the reduction of the propagation distance.
The new coordinate $y$ is more favorable for finding a unique projection of the ray manifold that disentangles the multipath
propagation. Still, this coordinate is not the best choice.

A much better coordinate for the new representation should be the impact parameter $p$, because in a spherically-symmetric
medium it is an invariant for each ray due to the Bouger law, and thus it is unique for each ray. A dynamic equation for the
variation of $p$ along the ray as a function of the horizontal gradient of refractivity was obtained by Gorbunov and Kornblueh
(2001). The idea of complementing the BP technique with one more transform that maps the field to the impact parameter
representation was pioneered by Gorbunov (2002). It was the first application of the FIO of the first type, which is linked to
the other type of the generating function (Arnold, 1978; Goldstein et al., 2014) and has the form

$$\hat{u}(p) = \sqrt{-\frac{ik}{2\pi}} \int a_1(p,\sigma)\exp(ikS_1(p,\sigma))\,\tilde{u}(\sigma)\,d\sigma \equiv \Phi_1[u(t)](p), \qquad (11)$$

where the only difference with the second type operator is that it acts upon the Fourier-transformed field $\tilde{u}(\xi)$. It can be looked
at as the composition of the Fourier transform, which itself is a second type FIO, and the other second type FIO. Because the
Fourier transform is a simple rotation of the phase space by $\pi/2$: $(t,\sigma)\to(\sigma,-t)$, the equation for the phase function takes the
form:

$$dS_1 = \xi dp + td\sigma. \qquad (12)$$

Gorbunov (2002) applied this operator to the back-propagated field. To this end, using the normal vector $\nu = \left(\eta,\sqrt{1-\eta^2}\right)$ to
the straight ray, we express the impact parameter:

$$p = -x\eta + y\sqrt{1-\eta^2}. \qquad (13)$$





Now it is necessary to find the canonical transform $(y, \eta) \to (p, \xi)$. We look for the first type operator, apply the property of

2-D canonical transforms that conserve the volume element, which follows from Eq. (12):

$$\frac{\partial \xi}{\partial \eta} \frac{\partial p}{\partial y} - \frac{\partial \xi}{\partial y} \frac{\partial p}{\partial \eta} = 1, \tag{14}$$

and additional assumption $\xi = \xi(\eta)$. Then, from Eq. 14, we readily derive:

$$\frac{\partial \xi}{\partial \eta} = \left(\frac{\partial p}{\partial y}\right)^{-1} = \frac{1}{\sqrt{1-\eta^2}},$$

$$\xi = \arcsin \eta \tag{15}$$

This results in the solution for the phase and amplitude functions:

$$S_2(p, \eta) = p \, \arcsin \eta - x\sqrt{1-\eta^2},$$

$$a_2(p, \eta) = \sqrt{\frac{\partial^2 S}{\partial p \partial \eta}} = \left(1-\eta^2\right)^{-1/4}. \tag{16}$$

This defines the FIO, which is applied to the backpropagated wave field $u_B(y)$ and produces the mapped field

$$\hat{u}(p) = A'(p) \exp\left(ik \int \xi(p) \, dp\right). \tag{17}$$

The derivative $\xi(p)$ of its eikonal is algebraically linked to the bending angle:

$$\epsilon(p) = -\xi(p - \arcsin\left(\frac{x_T p + y_T \sqrt{r_T^2 - p^2}}{r_T^2}\right), \tag{18}$$

where $(x_T, y_T) = \mathbf{r}_T$ is the transmitter position in the occultation plane. Because the cross-term in $S_2$, which depends both on $p$ and $\eta$, is linear with respect to $p$, the integration over new coordinate $\xi = \arcsin \eta$ turns it to $p\xi$ and, therefore, the operator is reduced to the Fourier transform in combination with a non-linear change of coordinate. This indicates that this operator

allows a fast implementation. A similar idea will be applied below.

The complicated nature of the BP+CT algorithm stimulated further studies (Gorbunov and Lauritsen, 2002, 2004b) where the idea was expressed of applying the FIO directly to the observed wave field $u(t)$, without intermediate and numerically expensive steps like BP. Full-Spectrum Inversion (FSI) developed by Jensen et al. (2003) was the first solution of this type, although with some restrictive assumptions. However, the general solution was just one year away: the Phase Matching (PM)

was developed by Jensen et al. (2004) and then put in the context of the CT approach by Gorbunov and Lauritsen (2004a), who also introduced an approach based on the linearized canonical transform that reduced the FIO to the composition of non-linear coordinate changes and Fourier transform. This algorithm was termed the 2nd type CT, or CT2.

In order to arrive at the phase function of the FIO of the 2nd type, consider the expression for the derivative of the phase of the observed wave field:

$$\dot{\Psi} = \sigma(p, y) = p\dot{\theta} + \frac{\dot{r}_T}{r_T}\sqrt{r_T^2 - p^2} + \frac{\dot{r}_R}{r_R}\sqrt{r_R^2 - p^2}, \tag{19}$$





Using Eq. (8), we derive the phase function:

$$
\begin{aligned}
S_2\left(p,t\right) &= -\int \left( p\dot\theta + \frac{\dot r_T}{r_T}\sqrt{r_T^2 - p^2} + \frac{\dot r_R}{r_R}\sqrt{r_R^2 - p^2} \right) dy = \\
&= -\int \left( p\,d\theta + \frac{dr_T}{r_T}\sqrt{r_T^2 - p^2} + \frac{dr_R}{r_R}\sqrt{r_R^2 - p^2} \right) = \\
&= -p\theta - \sqrt{r_T^2 - p^2} + p\arccos\frac{p}{r_T} - \sqrt{r_R^2 - p^2} + p\arccos\frac{p}{r_R},
\end{aligned}
\tag{20}
$$

where $\theta$, $r_T$, and $r_R$ are functions of time $t$. We don't reproduce here the derivation of the amplitude function $a_2\left(p,t\right)$, which uses simple geometrical considerations [Gorbunov2004b]. This phase function, although providing the accurate solution, has a disadvantage: its cross-term depending on both $p$ and $t$ is, generally speaking, not reduced to a form of $g_1\left(p\right)g_2(t)$. The FIO, in the generic case, cannot be reduced to a Fourier transform in composition with non-linear coordinate changes. This is, however, possible in the particular case of circular orbits, when the phase function equals $p\theta$, and using $\theta$ as a new coordinate instead of time reduces the operator to the Fourier transform. This method was referred to as FSI.

To find an approximate solution that significantly reduces the computational costs at an expense of an insignificant reduction of accuracy, the representation of the approximate impact parameter was introduced. The impact parameter $p$ is a function of $t,\sigma$: $p = p(t,\sigma)$. We introduce its approximation $\tilde p$:

$$
\tilde p\left(t,\sigma\right) = p_0\left(t\right) + \frac{\partial p_0}{\partial\sigma}\left(\sigma - \sigma_0\left(t\right)\right) = f\left(t\right) + \frac{\partial p_0}{\partial\sigma}\sigma,
$$

$$
f\left(t\right) = p_0\left(t\right) - \frac{\partial p_0}{\partial\sigma}\sigma_0\left(t\right) =
$$

$$
= p_0 - \left( \frac{d\theta}{dt} - \frac{dr_G}{dt}\frac{p_0}{r_G\sqrt{r_G^2 - p_0^2}} - \frac{dr_L}{dt}\frac{p_0}{r_L\sqrt{r_L^2 - p_0^2}} \right)^{-1}\sigma_0,
\tag{21}
$$

where $\sigma_0(t)$ is a smooth model of normalized Doppler frequency, $p_0(t) = p(t,\sigma_0(t))$, and $\partial p_0/\partial\sigma = \partial p/\partial\sigma|_{\sigma=\sigma_0(t)}$. We now parameterize the trajectory with the coordinate $\Upsilon = \Upsilon(t)$. For brevity we use the notation $u(\Upsilon)$ instead of $u(t(\Upsilon))$. For the coordinate $\Upsilon$ and the corresponding momentum $\eta$ we use the following definitions:

$$
d\Upsilon = \left(\frac{\partial p_0}{\partial\sigma}\right)^{-1} dt = \frac{\partial\sigma}{\partial p_0}dt,
$$

$$
\eta = \frac{\partial p_0}{\partial\sigma}\sigma.
\tag{22}
$$

Finally, we arrive at the following linear canonical transform $(\Upsilon,\eta) \to (p,\xi)$:

$$
\tilde p = f(\Upsilon) + \eta,
$$

$$
\xi = -\Upsilon,
\tag{23}
$$

The generating function of this canonical transform is easily computed from the differential equation

$$
dS_2 = \xi\,d\tilde p - \eta\,d\Upsilon = -\Upsilon\,d\tilde p - \left(\tilde p - f(\Upsilon)\right)d\Upsilon
$$

$$
S_2(\tilde p, Y) = -\tilde p Y + \int_0^Y f(Y')dY'.
\tag{24}
$$





For the new coordinate $\Upsilon$ we have the following relation:

$$d\Upsilon = d\theta - \frac{dr_G}{r_G}\frac{p_0}{\sqrt{r_G^2 - p_0^2}} - \frac{dr_L}{r_L}\frac{p_0}{\sqrt{r_L^2 - p_0^2}}. \tag{25}$$

For circular orbits, this approximation, once again, reduces to FSI. To evaluate the bending angle, we use the fact that the momentum of the field in the mapped space equals $-\Upsilon$. We also evaluate the accurate impact parameter $p$ as follows. Given the dependence $\Upsilon(\tilde{p})$, it possible to find the corresponding time $t(\tilde{p})$. Using Eq. (21), we infer:

$$\sigma(\tilde{p}) = (\tilde{p} - p_0(t(\tilde{p})))\left(\frac{\partial p_0}{\partial \sigma}\right)^{-1} + \sigma_0(t(\tilde{p})),$$

$$p(\tilde{p}) = p(t(\tilde{p}), \sigma(\tilde{p})). \tag{26}$$

Finally, for each impact parameter $p$, we determine the coordinate $\Upsilon(p) = -\xi(p)$ and, therefore, the corresponding moment of time $t = t(\Upsilon(p))$, when this ray was observed, the bending angle is then evaluated from the geometrical relation:

$$\epsilon(p) = \theta(t(\Upsilon(p))) - \arccos\frac{p}{r_T(t(\Upsilon(p)))} - \arccos\frac{p}{r_R(t(\Upsilon(p)))}. \tag{27}$$

This method termed CT2 indicates both a high accuracy and numerical performance. This discourse leads us to the conclusion that there is a family of closely related WO methods that are based on the same principle. The observed wave field is subjected to

a linear integral operator with an oscillating kernel that transforms the field into a different representation. The representation is chosen in such a way that the projection of the ray manifold to the new coordinate axis is unique. The operation is also referred to as unfolding multipath. Finally, such methods as CT, FSI, PM, and CT2 involve the evaluation of the same integral transform under different assumptions and approximations. The difference in the results of the application of these WO methods is less significant than the difference coming from other parts of RO data processing systems, including cut-off, filtering, and quality

control procedures.

## 3.2 Generalized Canonical Transform method

All the modifications of the CT approach discussed above relied upon impact parameter $p$ as the unique coordinate of the ray manifold. However, impact parameter is, generally speaking, not invariant for each ray, and its perturbations due to horizontal gradients may result in breaking the above condition. To see this, consider the ray equations in the Hamilton form. The are

derived from the Hamilton function:

$$H(\mathbf{r}, \mathbf{p}) = \frac{1}{2}\left(\mathbf{p}^2 - n^2(\mathbf{r})\right), \tag{28}$$

where $\mathbf{p}$ is the momentum, and $n(\mathbf{r})$ is the refractivity field. The Hamilton system has the following form:

$$\dot{\mathbf{r}} = \frac{\partial H}{\partial \mathbf{p}}, \quad \dot{\mathbf{p}} = -\frac{\partial H}{\partial \mathbf{r}}, \quad \dot{\Psi} = \mathbf{p}\dot{\mathbf{r}},$$

$$\dot{\mathbf{r}} = \mathbf{p}, \quad \dot{\mathbf{p}} = n\nabla n, \quad \dot{\Psi} = n^2, \tag{29}$$





where $\mathbf{p}$ is the classical momentum. Because $|\mathbf{p}| = |\nabla\Psi| = n$, we arrive at the following differential relation between the parameter $\tau$ of this system, the ray arc length $s$, and the eikonal:

$$d\tau = \frac{ds}{n}, \quad d\Psi = n \, ds. \tag{30}$$

Equation (29) has a form that is specific for the Cartesian coordinates. Consider an arbitrary coordinate system with the metric tensor $g_{ij}$: $ds^2 = dx^i g_{ij} dx^j$, where $x^i$ are the components of vector $\mathbf{r}$, and we follow the Einstein tensor notation implying the

summation over each pair of upper and lower indexes of the same name. If we define the momentum by the relation $p_i = g_{ij}\dot{x}^j$, the form $\mathbf{p} \, d\mathbf{r}$ is invariant, the transform to the new coordinates $(p^i, \, x^i)$ is canonical, and the canonical form of the Hamilton system also remains invariant [Arnold1978], provided that the Hamilton function is defined as follows:

$$H(\mathbf{r}, \mathbf{p}) = \frac{1}{2}\left(p_i g^{ij} p_j - n^2(\mathbf{r})\right), \tag{31}$$

where $g^{ij}$ is the matrix inverse to $g_{ij}$. This results in the following form of the ray equations:

$\dot{x}^i = \dfrac{\partial H}{\partial p_i} = g^{ij}p_j, \dot{p}_i = -\dfrac{\partial H}{\partial x^i} = n\dfrac{\partial n}{\partial x^i} - \dfrac{1}{2}p_k\dfrac{\partial g^{kj}}{\partial x^i}p_j.$

The 2-D approximation (Zou et al., 2002) allows treating rays as plane curves. Consider polar coordinates $(r, \theta)$ with the metric tensor:

$$g_{ij} = \begin{pmatrix} 1 & 0 \\ 0 & r^2 \end{pmatrix}, \quad g^{ij} = \begin{pmatrix} 1 & 0 \\ 0 & r^{-2} \end{pmatrix}. \tag{32}$$

Then we have the following equations:

$p_\theta = r^2\dot{\theta} = nr\dfrac{rd\theta}{ds} = nr\sin\psi,$

$\dot{p}_\theta = n\dfrac{\partial n}{\partial \theta},$

$\dot{p}_r = \ddot{r} = n\dfrac{\partial n}{\partial r} + \dfrac{p^2}{r^3}. \tag{33}$

where $\psi$ is the angle between vectors $\dot{\mathbf{r}}$ and $\mathbf{r}$. The angular component of the momentum $p_\theta$ coincides with the ray impact parameter $p$, which is invariant in a spherically layered medium, but is perturbed by the horizontal gradients (Gorbunov and Lauritsen,

295    2009).

     The variations of the ray impact parameter seem to undermine the elegant idea of the CT approach. Now there is no such a convenient invariant value ascribed to each ray. Still, the CT method can be applied using the same formulas, but the coordinate $p$ will now acquire a different meaning: it will be understood as the "effective impact parameter", i.e. the impact parameter which would result in the observed Doppler frequency shift, if the atmosphere were spherically layered [Gorbunov2019].

Accordingly, the evaluated bending angle will also be the "effective" bending angle. The reason is that for the evaluation of the real bending angle, understood as the angle between the ray directions at the transmitter and receiver, two corresponding values of the impact parameter are required, which cannot be derived from the single variable, the Doppler frequency. This,

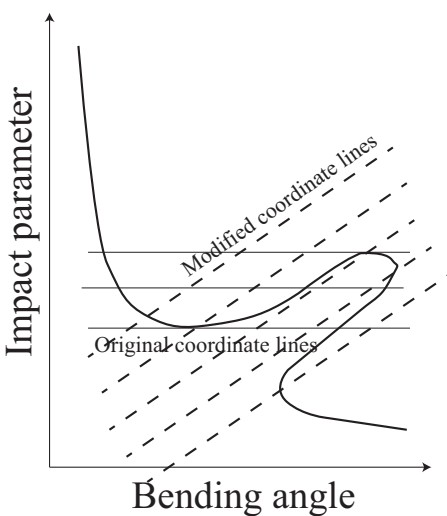

**Figure 2.** Impact multipath, old coordinate (impact parameter) lines, and modified coordinate lines.

by itself, is not a significant problem, because the assimilation of bending angle profiles can be based on the effective values [Gorbunov2019], provided that the observation operator correctly implements their evaluation.

More importantly, horizontal gradients may result in multi-valued ray manifold projections, when using the effective impact parameter $p$ as the coordinate in the mapped space. This situation is termed "impact multipath" [Zou2019]. Theoretically, for any ray manifold perturbation there always exists an unfolding coordinate transform. This follows from the fact that topologically the ray manifold is always a continuous line without self-crossing. However, this coordinate transform will now depend on the a priori unknown horizontal gradients of refractivity.

Typical multi-valued bending angle profile [Gorbunov2009a, Zou2019] is shown in Figure 2. From numerical simulations, it can be inferred that there is a kind of asymmetry: impact multipath manifests itself mostly in ascending spikes, but hardly in descending spikes. Accodingly, in order to better unfold multipath, it must be possible to use another coordinate in such a way that the modified coordinate lines are sloped. Therefore, we modify the transform (23) in order to use another coordinate:

$$\widetilde{p'} = \tilde{p} + \beta\Upsilon, \tag{34}$$

where $\beta$ is a tunable parameter and has a dimension of km/rad. Although the optimal value of this parameter should be different for individual events, the aforementioned asymmetry results in the conclusion that the preferred value of $\beta$ is expected to be negative. Therefore, it may be possible to find its optimal value that, in the statistical sense, will minimize errors due to impact multipath.

The modified canonical transform (23) is written as follows:

$$\widetilde{p'} = f(\Upsilon) + \beta\Upsilon + \eta \equiv f'(\Upsilon) + \eta,$$
$$\xi = -\Upsilon. \tag{35}$$



Using the modified function $f'(\Upsilon)$ instead of the original one, we will obtain the expression for the modified FIO $\widehat{\Phi'}_2$. The advantage of this approach is that it can be implemented by a very simple modification of the existing CT2 algorithm. Using the numerical implementation of the modified CT will allow us to study its influence upon the RO inversion statistics in the lower troposphere.

Denote the generalized FIO $\widehat{\Phi}_2^{(\beta)} u(\tilde{p})$ Consider the wave field in the impact parameter representation, $\hat{u}\left(\beta;\tilde{p}'\right) = \widehat{\Phi}_2^{(\beta)} u\left(\tilde{p}'\right)$. The standard CT algorithm corresponds to the evaluation of $\hat{u}(0;\tilde{p}) = \widehat{\Phi}_2^{(0)} u(\tilde{p})$ with $\beta = 0$.

It is possible to arrive at a quantitative estimate of $\beta$ based on (Gorbunov and Kornblueh, 2001; Gorbunov and Lauritsen, 2009; Zou et al., 2019). We expect that $|\beta| \lesssim \delta p/\delta\epsilon$, where $\delta p$ is the typical variation of impact parameter due to the horizontal gradients, and $\delta\epsilon$ is the corresponding bending angle variation. Assuming that $\delta p \approx 0.1$ km, and $\delta\epsilon \approx 0.01$ rad, we arrive at to arrive at a first quantitative estimate of $\beta \approx -10$ km/rad.

### 3.3 Affine transform for updating existing CT algorithms

Modification of existing numerical algorithms may not be so straightforward, as it follows from the above mathematical considerations. In order to avoid this, it is possible to complement an existing implementation of any WO-based numerical algorithm by an additional affine transform.

We will now derive the transform between $\hat{u}(0;\tilde{p})$ and $\hat{u}\left(\beta;\tilde{p}'\right)$. We can write the following transform between these representations:

$$\tilde{p}' = \tilde{p} - \beta(\xi - \xi_0),$$
$$\xi' = \xi, \tag{36}$$

where $\xi_0$ is the reference point. This is an affine transform in the $(\tilde{p}, \xi)$ plane. This suggests the abbreviation CT2A for the new generalized form, which stands for the CT2 complemented with the affine transform.

The generating function of transform (36) $S\left(\tilde{p}',\xi\right)$ is defined by

$$dS^{(\beta)} = \xi d\tilde{p}' + \tilde{p}d\xi, \tag{37}$$

which is equivalent to the following system:

$$\frac{\partial S^{(\beta)}}{\partial \tilde{p}'} = \xi,$$

$$\frac{\partial S^{(\beta)}}{\partial \xi} = \tilde{p} = \tilde{p}' + \xi(\xi - \xi_0). \tag{38}$$

From this, we can conclude that

$$S^{(\beta)}\left(\tilde{p}',\beta\right) = \tilde{p}'(\xi - \xi_0) - \beta\frac{(\xi - \xi_0)^2}{2}. \tag{39}$$

This phase function defines the FIO of the first type:

$$\hat{u}\left(\beta;\tilde{p}'\right) = \sqrt{-\frac{ik}{2\pi}} \int \exp\left(ikS^{(\beta)}\left(\tilde{p}',\xi\right)\right)\tilde{u}(\xi)\,d\xi \equiv \widehat{\Phi}_1^{'(\beta)}[u(t)](\tilde{p}) \tag{40}$$





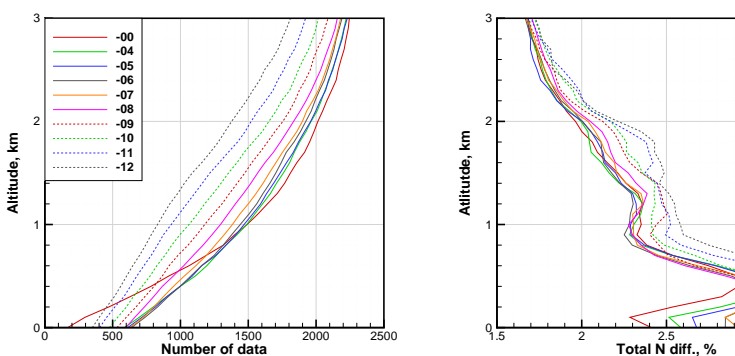

**Figure 3.** Statistics for latitude band $0°$–$10°$. Left: number of data; right total relative difference of refractivity COSMIC–ECMWF $\sqrt{\langle (N_C - N_E)^2 \rangle}/N_E$. Both are functions of the parameter $\beta$.

Finally, we can write the operator relation:

$$\widehat{\Phi}_2^{(\beta)} = \widehat{\Phi}_1^{'(\beta)} \widehat{\Phi}_2^{(0)}, \tag{41}$$

which can be used for the modification of the existing version of operator $\widehat{\Phi}_2^{(0)}$.

The above derivation allows for one more generalization. We can consider $\beta = \beta(\xi)$. In this case, the phase function is
derived in a straightforward way:

$$S^{(\beta)}\left(\tilde{p}', \xi\right) = \tilde{p}'(\xi - \xi_0) - \int \beta(\xi)(\xi - \xi_0)\,d\xi. \tag{42}$$

Using $\beta(\xi) = \sum \beta_j \xi^j$ results in a simple analytical expression for $S^{(\beta)}$ with a set of tuning parameters $\beta_j$. In this work, we, however, use a constant $\beta$.

## 4   Implementation and numerical performance evaluation

Our implementation of the CT2A algorithm was based on the existing program code with addition of the parameter $\beta$ and using the modified function $f'(\Upsilon)$ as defined by Eq. (35). Practically, this only required modification of a few lines in the program code that implements the CT2 method, as well as the implementation of one more command line parameter.

In our numerical validation, we retrieved COSMIC refractivity profiles $N_C$, using COSMIC data from the year 2008, 1st and 15th day of every month, leading to a total of 24 days and altogether around 60000 RO events. We used collocated ECMWF
refractivity profiles $N_E$, i.e., interpolated to the corresponding COSMIC RO event location, as the reference. We used the total relative difference of COSMIC from ECMWF (the difference metric), defined as $\sqrt{\langle (N_C - N_E)^2 \rangle}/N_E$, which includes both mean (systematic) and fluctuating (random) deviations.




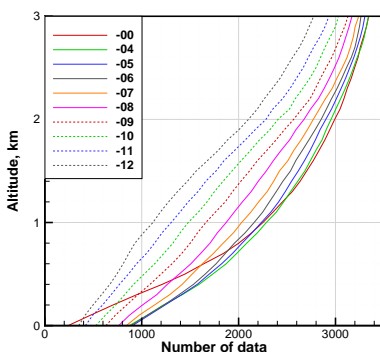
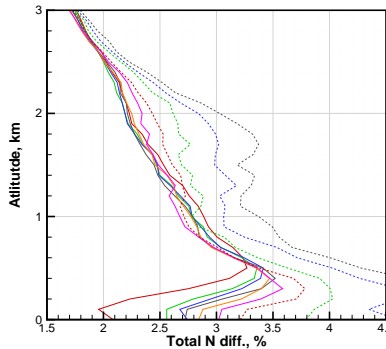

**Figure 4.** Statistics for latitude band $10°–20°$. Left: number of data; right total relative difference of refractivity COSMIC–ECMWF $\sqrt{\left\langle (N_C - N_E)^2 \right\rangle}/N_E$. Both are functions of the parameter $\beta$.

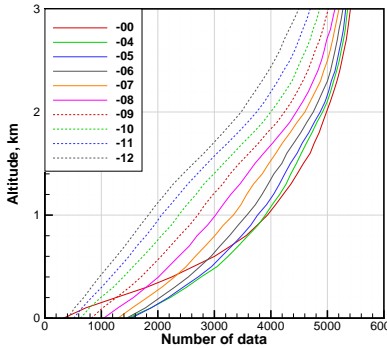
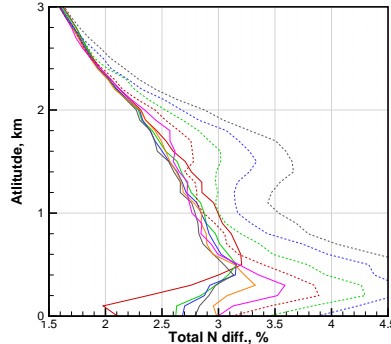

**Figure 5.** Statistics for latitude band $20°–30°$. Left: number of data; right total relative difference of refractivity COSMIC–ECMWF $\sqrt{\left\langle (N_C - N_E)^2 \right\rangle}/N_E$. Both are functions of the parameter $\beta$.

Figure 3 through 11 show the statistical values of $\sqrt{\left\langle (N_C - N_E)^2 \right\rangle}/N_E$ as function of latitude and parameter $\beta$. We averaged over $10°$ wide latitude bands including both South and North hemispheres. The parameter $\beta$ changed in the interval
from $-4$ to $-12$ km/rad with the step of 1.

These results indicate that for latitudes $0°–50°$, in the altitude range from 0.5 km to 1.9–2.5 km, the application of the CT2A algorithm allows minimizing the total relative difference of refractivity profiles COSMIC–ECMWF $\sqrt{\left\langle (N_C - N_E)^2 \right\rangle}/N_E$. The optimal value of parameter $\beta$ is found to be $-6$ to $-8$ km/rad. The CT2A algorithm also improves the penetration increasing the number of data in the altitude range below 0.5 km. Here the difference metrics for $\beta = 0$ and optimal $\beta$ cannot be
directly compared, because they are evaluated over different statistical ensembles.





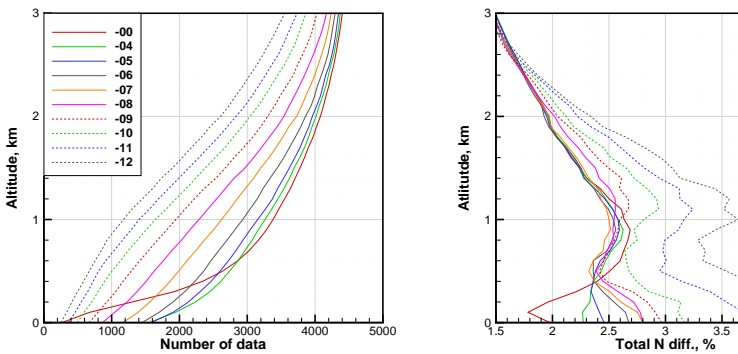

**Figure 6.** Statistics for latitude band $30°$–$40°$. Left: number of data; right total relative difference of refractivity COSMIC–ECMWF $\sqrt{\langle (N_C - N_E)^2 \rangle}/N_E$. Both are functions of the parameter $\beta$.

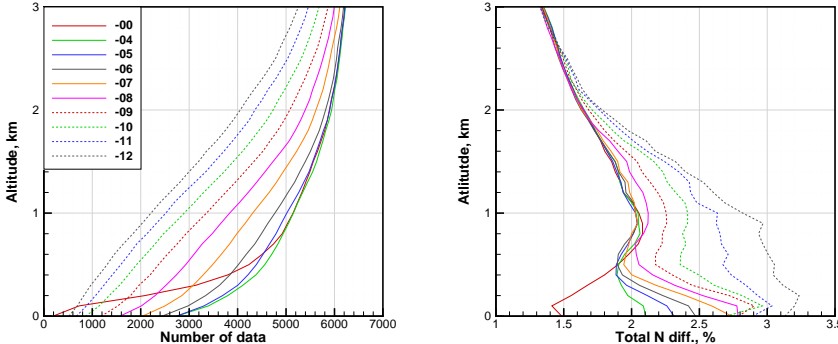

**Figure 7.** Statistics for latitude band $40°$–$50°$. Left: number of data; right total relative difference of refractivity COSMIC–ECMWF $\sqrt{\langle (N_C - N_E)^2 \rangle}/N_E$. Both are functions of the parameter $\beta$.

## 5 Summary and conclusions

In this study we discussed the general idea of the Canonical Transform (CT) method and provided a new generalization adding more flexibility for application in RO processing. The idea came from quantum mechanics, where it was shown that the canonical transforms as they are understood in classical mechanics (geometrical optics) are implemented in quantum mechanics

(wave optics) by linear operators with oscillating kernels. Such operators are referred to as Fourier Integral Operators (FIOs). During the past century, this approach acquired a solid theoretical basis. In numerous mathematical monographs, one finds the advanced theory of FIOs. The central role in this theory is played by the concept of the ray manifold and its projections.





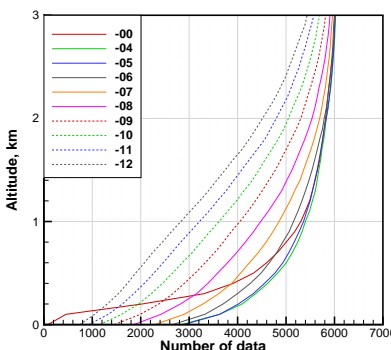
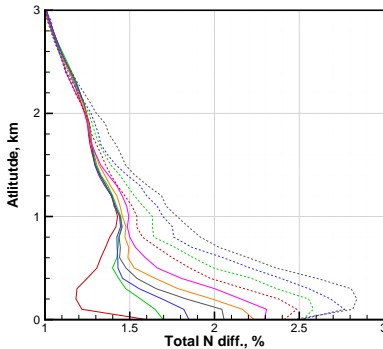

**Figure 8.** Statistics for latitude band $50°$–$60°$. Left: number of data; right total relative difference of refractivity COSMIC–ECMWF $\sqrt{\langle (N_C - N_E)^2 \rangle}/N_E$. Both are functions of the parameter $\beta$.

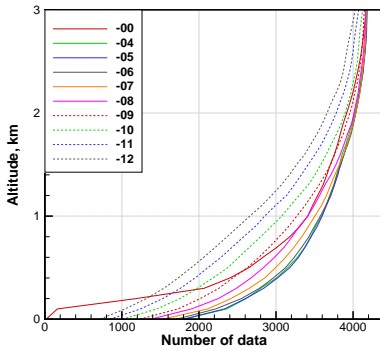
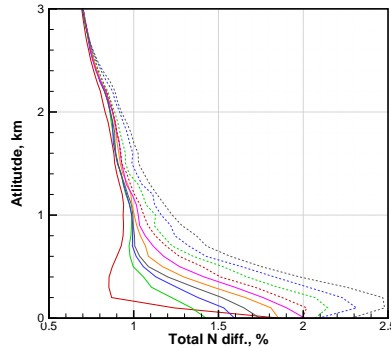

**Figure 9.** Statistics for latitude band $60°$–$70°$. Left: number of data; right total relative difference of refractivity COSMIC–ECMWF $\sqrt{\langle (N_C - N_E)^2 \rangle}/N_E$. Both are functions of the parameter $\beta$.

The CT method has been applied for RO observations for a long time. Although there have been many modifications, like original CT combined with Back Propagation (BP), Full-Spectrum Inversion (FSI), Phase Matching (PM), and CT of type 2 (CT2), there is no essential difference between these FIO-based methods. The difference consists in the approximation of the phase function of the FIO, leading to the corresponding approximate representation of the impact parameter and bending angle, and in the specific implementation (such as cut-off, filtering, and quality control procedures). All these methods map the wave field to the representation of the impact parameter $p$. The reason for this choice of the coordinate in the mapped space is that in the case of a spherically-symmetric medium, the impact parameter is always a unique coordinate of the ray manifold.

Because the real atmosphere is not spherically-symmetric, this results in some aggravation. First, in the strict sense, there is no such a quantity as the impact parameter as a unique variable any more. But it is still possible to operate with the effective





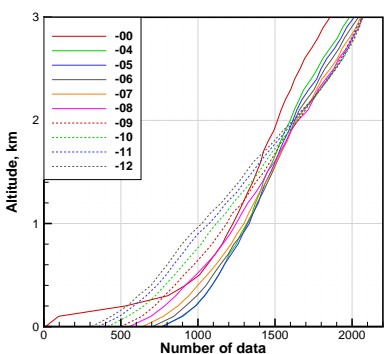
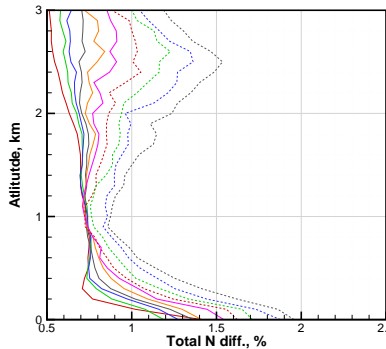

**Figure 10.** Statistics for latitude band $70°$–$80°$. Left: number of data; right total relative difference of refractivity COSMIC–ECMWF $\sqrt{\langle (N_C - N_E)^2 \rangle}/N_E$. Both are functions of the parameter $\beta$.

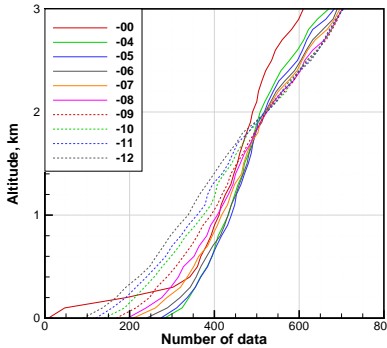
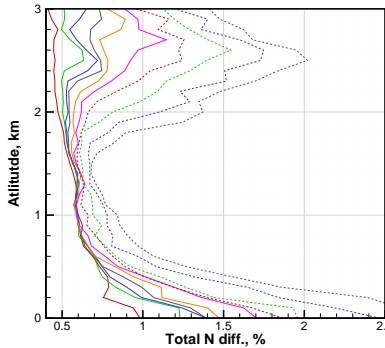

**Figure 11.** Statistics for latitude band $80°$–$90°$. Left: number of data; right total relative difference of refractivity COSMIC–ECMWF $\sqrt{\langle (N_C - N_E)^2 \rangle}/N_E$. Both are functions of the parameter $\beta$.

impact parameter, derived from Doppler frequency shift using the same relations as for a spherically-symmetric medium. This quantity can be implemented in the observation operator for the variational assimilation of RO observation, cancelling errors due horizontal gradients. However, the above property of the impact parameter, which is supposed to be the unique coordinate

of the ray manifold, does not always hold for the effective value. In some cases, the situation referred to as the impact multipath may occur, resulting in retrieval errors in atmospheric profiles derived from RO data.

In order to partially mitigate this fundamental shortcoming, we introduced a generalization of the CT approach. We used a generalized definition of the coordinate in phase space, defined as a linear combination of impact parameter and bending angle. Because this can be understood as an affine transform of the phase space, we coined the abbreviation CT2A for the new

method. This transform has a parameter $\beta$, which can be tuned to minimize the retrieval error.



To find such a value of the parameter by statistical performance evaluation under real RO observation conditions including challenging horizontal gradients in the lower troposphere, we processed a large ensemble of COSMIC RO data for the year 2008, 1st and 15th day of every month, adding up to a total of about 60000 RO events. We used the total relative difference of COSMIC from collocated ECMWF analysis profiles over the lower troposphere as the metric for this evaluation and the tuning
parameter estimation.

For latitudes $0°–50°$, in the altitude range from 0.5 km to 1.9–2.5 km, the application of the CT2A algorithm was used to statistically minimize the COSMC–ECMWF difference metric and the optimal value of parameter $\beta$ is found to be $-6$ to $-8$. We found that the CT2A algorithm as well improves the penetration statistics of RO profile retrievals, increasing the number of data in the altitude range below 0.5 km.

Overall these results suggest that the CT2A method is not only theoretically an innovative generalization of the CT/FIO class of methods but also practically a valuable advancement for RO processing in that it can improve the capability to cope with challenging horizontal gradient conditions in the lower troposphere.

*Author contributions.* K.B.Lauritsen: The problem formulation, the initial idea of the study, theoretical discussion, contribution to finalizing the manuscript; M.Gorbunov: Theoretical derivations, numerical implementation, statistical study, initial draft of the paper; G.Kirchengast:
theoretical discussions, contribution to finalizing the manuscript

*Competing interests.* The authors declare that no competing interests are present.

*Acknowledgements.* M.E. Gorbunov is grateful to Russian Foundation for Basic Research (grant No 20-05-00189 A) for the financial support. G. Kirchengast acknowledges support, including for partial co-funding of the work of M.E. Gorbunov, by the Aeronautics and Space Agency of the Austrian Research Promotion Agency (FFG-ALR) under the Austrian Space Applications Programme (ASAP) project ATROMSAF1
(proj.no. 859771) funded by the Ministry for Transport, Innovation, and Technology (BMVIT). K.B. Lauritsen has been supported by the Radio Occultation Meteorology Satellite Application Facility (ROM SAF) which is a decentralized operational RO processing center under EUMETSAT.





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
