# Peer review of "Generalized Canonical Transform method for radio occultation sounding with improved retrieval in the presence of horizontal gradients"

_Atmospheric Measurement Techniques, 2020_

## Referee Comment (RC1) · Anonymous Referee #1 · 11 Jun 2020

The derivation of atmospheric refractivity profiles from GNSS radio occultation (RO) observations in the lower and mid troposphere is a challenging task for several reasons, the most important being multipath ray propagation which leads to signal interference effects at the receiver location. To address multipath the present authors over the last two decades have developed bending angle retrieval techniques based on transformations ("canonical transformations") from geometric to impact parameter space. These wave optical techniques, the most advanced being the CT2 algorithm, are based on the assumption, that the atmospheric refractivity field is spherically symmetric. Validation studies on the performance of the wave optical approaches confirmed their superior performance in the mid troposphere when compared to the geometric optics method.

[Figure]

In the lower troposphere, however, horizontal gradients in the refractivity field induce deviations from spherical symmetry, in particular at low latitudes, and the retrieved refractivity profiles may differ by a few percent from corresponding ECMWF data.

In the present study the authors address the lack of spherical symmetry in the lower tropospheric refractivity field by introducing a linear coordinate transformation in impact parameter space such that bending angle is a single-valued function of a generalized "impact parameter". The key feature of this approach ("CT2A") is the introduction of an a-priori unknown parameter $\beta$, which needs to be adjusted.

The first two sections of this paper provide an excellent summary of the development from the original "back propagation" to the current CT2 method. The third section describes the generalization of the "canonical transform" method. The introduction of an adjustable parameter to the wave optical retrieval is conceptually a significant step forward, even if the benefit of the CT2A algorithm to the refractivity retrieval is not immediately obvious. At low latitudes the CT2A method appears to improve the data yield within the planetary boundary layer, albeit at the expense of larger deviations from ECMWF. At polar latitudes horizontal gradients are expected to be less pronounced and therefore the CT2A algorithm does not perform better than CT2. In summary, I consider this paper a valuable contribution to the topic of RO retrievals in the lower troposphere and I recommend its publication.

The latitudinal dependence of the fractional refractivity deviation on the choice of the parameter $\beta$ is an intriguing feature of the present analysis. In the subtropics at altitudes between 1 and 2 km the sensitivity (in particular for $\beta < -10$ km/rad) is larger compared to the latitude band $-10°$S to $10°$N. An obvious question is if this latitudinal dependence is correlated with the latitudinal dependence of strong (horizontal) refractivity gradients extracted from ECMWF meteorological fields. Second, I would suggest to improve the graphical representation of the results by splitting the right panel in Figs. 3 to 11 into two panels, one showing the CT2 results ($\beta = 0$ km/rad) and the other the *difference* between CT2A and CT2.

Technical corrections:

Page 2, line 46:
"a short-wave asymptotical solution" → "a short-wave asymptotic solution"

Page 2, line 51:
Gorbunov et al. (2004) probably should read Gorbunov and Lauritsen (2004b).

Page 3, line 69:
"which is known the Bouger law" → "which is known as Bouguer's law"

Page 4, line 117:
"although it may have multiple projections to the axis of time $t$,"
I would suggest instead "although it may not be single-valued with respect to time $t$," or similar.

Page 5, line 138:
"instant frequency" → "instantaneous frequency"

Page 7, line 173:
"This transform is performed under the application of the procedure of the stationarization of the transmitting satellite [...]"
I suggest to rephrase this sentence.

Page 7, line 186:
I assume here $\tilde{u}(\xi)$ should read $\tilde{u}(\sigma)$ instead.

Page 8, eqn. 18:
A closing bracket is missing.

Page 9, line 226:
[Gorbunov2004b] probably should read (Gorbunov and Lauritsen, 2004b).

Page 10, line 263:
"The difference in the results of the application of these WO methods is less significant

than the difference coming from other parts of RO data processing systems, [...]"
I suggest to add a reference.

Page 11, line 282:
[Arnold1978] → (Arnold, 1978)

Page 11, line 299 and page 12, line 304:
[Gorbunov2019] is not listed in the reference section.

Page 19, line 407:
"COSMC–ECMWF" → "COSMIC–ECMWF"

Page 21, line 476 and 478:
"Intoduction" → "Introduction"

―――――――――――――――――――――

---

## Referee Comment (RC2) · Anonymous Referee #2 · 29 Aug 2020

Generalized Canonical Transform method for radio occultation sounding with improved retrieval in the presence of horizontal gradients

M. Gorbunov, G. Kirchengast and K. Lauritsen

General comments

The authors present a new wave optics retrieval method aimed at mitigating "systematic errors" (line 15) in the lower troposphere. The central problem being addressed is that GNSS radio occultation (GNSS-RO) wave optics retrievals rely on a co-ordinate transform from time and Doppler to impact parameter and bending angle. However, impact parameter is not necessarily unique in the presence of horizontal gradients. This is sometimes called "impact multipath". The authors suggest a new transform, "CT2A", combining both bending angles and impact parameters, which depends on a tuneable parameter, β. It is noted that the optimal value of β will vary for each individual measurement, but it may be possible to estimate a value that is optimal in a statistical sense. A range of plausible values are tested in the paper.

The paper does a good job reviewing of GNSS-RO wave optics developments (many of which the authors have led), and presenting them in a broader physical context. The citation of other work is also fair. I do not follow all the mathematical details of the new approach, but the physical reasoning seems correct. However, major revision is required before publication. A significant difficulty with this paper is understanding whether the new approach (β≠0) is better than the current method (β=0). For example, the authors say that (line 374):

"Here the difference metrics for β = 0 and optimal β cannot be directly compared, because they are evaluated over different statistical ensembles."

This makes the interpretation of the results presented in Figures 3-11 extremely difficult. The penetration depth alone does not seem to be a strong argument, particularly when β=0 often provides more data above 1 km. It would be more useful to show the subset of refractivity values common to all retrievals.

In addition, the text says that the method mitigates systematic errors, but the metric shown in these figures combines systematic and random errors. I suggest that the systematic and random error estimates should be plotted separately. These points and the specific comments given below should be addressed before publication.

Specific comments

Line 225: "don't" should be "do not".

Line 293: "The angular component of the momentum $p_\vartheta$ coincides with the ray impact parameter $p$, which is invariant in a spherically layered medium, but is perturbed by the horizontal gradients (Gorbunov and Lauritsen, 2009)". Healy (2001) also pointed this out.

Line 299: [Gorbunov2019] not listed in references. Format of reference in text.

The references appear to change format e.g., line 306 "[Zou2019]" and line 310 "[Gorbunov2009a, Zou2019]". These should be (Zou et al., 2019) and (Gorbunov and Lauritsen, 2009).

Line 364: "co-located ECMWF refractivity profiles". It would be useful to give more detail here. For example, does this computation include the tangent point drift? Do you compute the refractivity directly from the ECMWF P,T and Q fields. Are they ECMWF forecasts or analyses? What resolution?

Line 366: It would be useful to split this metric into to systematic and random errors instead of combining them, particularly if the transform is likely to improve systematic errors, as noted in the abstract.

Line 373: "The CT2A algorithm also improves the penetration increasing the number of data in the altitude range below 0.5 km."

This is correct, but $\beta = 0$ appears to provide more data above 1 km. Why is this? Are you using the transformed amplitude to cut-off the data? Please explain.

Line 374: "Here the difference metrics for $\beta = 0$ and optimal $\beta$ cannot be directly compared, because they are evaluated over different statistical ensembles.".

This really makes it difficult for the reader to judge whether the new transform is an advantage or not in all the subsequent figures. Is it possible to present the results for a dataset common to all $\beta$ values to help the reader interpret the results?

---

## Author Comment (AC2) · 2 Oct 2020

"Here the difference metrics for  $\beta = 0$  and optimal  $\beta$  cannot be directly compared, because they are evaluated over different statistical ensembles."

This makes the interpretation of the results presented in Figures 3-11 extremely difficult. The penetration depth alone does not seem to be a strong argument, particularly when  $\beta = 0$  often provides more data above 1 km. It would be more useful to show the subset of refractivity values common to all retrievals.

Following the suggestion of the Reviewer we performed some further study and found another important property of CT2A, as discussed below.

In addition, the text says that the method mitigates systematic errors, but the metric shown in these figures combines systematic and random errors. I suggest that the systematic and random error estimates should be plotted separately. These points and the specific comments given below should be addressed before publication.

The statement about the mitigation of the bias was made in a preliminary study, which was based on a much smaller volume of data. After the full study, we made a conclusion that it is the mean square difference between the RO and ECMWF refractivities that can be minimized by using the modified algorithm. Therefore, based on this additional finding, we refined the formulation in the abstract.

Line 225: "don't" should be "do not".

OK.

Line 293: "The angular component of the momentum pd coincides with the ray impact parameter p, which is invariant in a spherically layered medium, but is perturbed by the horizontal gradients (Gorbunov and Lauritsen, 2009)". Healy (2001) also pointed this out.

Healy (2001) refers to the technical report (Gorbunov, 1996), where the derivation of the impact parameter variation using the Hamiltonian form of ray trajectory equation was first presented.

Line 299: [Gorbunov2019] not listed in references. Format of reference in text. The references appear to change format e.g., line 306 "[Zou2019]" and line 310 "[Gorbunov2009a, Zou2019]". These should be (Zou et al., 2019) and (Gorbunov and Lauritsen, 2009). We corrected the references (cf. the similar remarks of Reviewer #1). That was related to technical corrections regarding the LATEX.

Line 364: "co-located ECMWF refractivity profiles". It would be useful to give more detail here. For example, does this computation include the tangent point drift? Do you compute the refractivity directly from the ECMWF P, T and Q fields? Are they ECMWF forecasts or analyses? What resolution?

We used ECMWF analyses at 1-degree latitudinal and 1-degree longitudinal resolution, with 91 vertical level covering the altitude range up to about 80 km. The refractivity was evaluated from pressure, temperature, and humidity fields. The tangent point drift was taken into account. We checked that this is also noted in the manuscript so that it is clear to the readers.

Line 366: It would be useful to split this metric into to systematic and random errors instead of combining them, particularly if the transform is likely to improve systematic errors, as noted in the abstract.

We preferred to correct the statement about the systematic errors.

Line 373: "The CT2A algorithm also improves the penetration increasing the number of data in the altitude range below 0.5 km." This is correct, but  $\beta = 0$  appears to provide more data above 1 km. Why is this? Are you using the transformed amplitude to cut-off the data? Please explain.

This is linked to the QC procedure and still needs further investigation that will be performed beyond the scope of this initial introduction study of the CT2A.

Line 374: "Here the difference metrics for  $\beta = 0$  and optimal  $\beta$  cannot be directly compared, because they are evaluated over different statistical ensembles.". This really makes it difficult for the reader to judge whether the new transform is an advantage or not in all the subsequent figures. Is it possible to present the results for a dataset common to all  $\beta$  values to help the reader interpret the results?

As noted above, we evaluated the statistics for the common dataset and found another important property of CT2A. The statistical differences between refractivity retrieved with  $\beta = 0$  and other values of  $\beta$  is vanishingly small (never exceeding a level of 0.0005%), but increasing  $\beta$  provide decreasing deviation from ECMWF and decreasing number of data. This indicates that CT2A allows the implementation of a QC procedure not involving any external data and only based on the internal properties of observed signals. This can be interpreted as follows. By extracting inversions that are common for different values of  $\beta$  we look at the ray manifold in the phase space from different directions and only choose events, where the ray manifold structure is stable. We modified the abstract and the respective parts of the text accordingly.

---

## Author Response (AR1)

*The latitudinal dependence of the fractional refractivity deviation on the choice of the parameter β is an intriguing feature of the present analysis. In the subtropics at altitudes between 1 and 2 km the sensitivity (in particular for 3 < −10 km/rad) is larger compared to the latitude band −10°S to 10°N. An obvious question is if this latitudinal dependence is correlated with the latitudinal dependence of strong (horizontal) refractivity gradients extracted from ECMWF meteorological fields.*

Yes, as visible from the manuscript, the latitudinal dependence is correlated with strong horizontal gradients of refractivity both in the real atmosphere and in ECMWF meteorological fields.

*Second, I would suggest to improve the graphical representation of the results by splitting the right panel in Figs. 3 to 11 into two panels, one showing the CT2 results (β = 0 km/rad) and the other the difference between CT2A and CT2.*

Ok, we agree this improves the visual presentation of the results. We hence added one more panel with the CT2A–CT2 difference.

**Technical corrections:**

*Page 2, line 46:*
*"a short-wave asymptotical solution" → "a short-wave asymptotic solution"*

OK.

*Page 2, line 51:*
*Gorbunov et al. (2004) probably should read Gorbunov and Lauritsen (2004b).*

OK.

*Page 3, line 69:*
*"which is known the Bouger law" → "which is known as Bouguer's law"*

OK.

*Page 4, line 117:*
*"although it may have multiple projections to the axis of time t,"*

*I would suggest instead "although it may not be single-valued with respect to time t," or similar.*

OK.

*Page 5, line 138:*
*"instant frequency" → "instantaneous frequency"*

OK.

*Page 7, line 173:*
*"This transform is performed under the application of the procedure of the stationarization of the transmitting satellite [...]" I suggest to rephrase this sentence.*

The sentence is rephrased as follows: "This transform is preceded by the stationarization…"

*Page 7, line 186:*
*I assume here $\tilde{u}(\xi)$ should read $\tilde{u}(\sigma)$ instead.*

Yes.

*Page 8, eqn. 18: A closing bracket is missing.*

OK.

*Page 9, line 226:*
*[Gorbunov2004b] probably should read (Gorbunov and Lauritsen, 2004b).*

Ok, rectified. We had missed the LATEX citation command here.

*Page 10, line 263:*
*"The difference in the results of the application of these WO methods is less significant than the difference coming from other parts of RO data processing systems, [...]" I suggest to add a reference.*

Ok, we added two references:

Gorbunov, M. E.; Shmakov, A. V.; Leroy, S. S. & Lauritsen, K. B. (2011), 'COSMIC Radio Occultation Processing: Cross-center Comparison and Validation', *J. Atmos. Oceanic Technol.* **28**(6), 737--751.

Gorbunov, M. E.; Benzon, H.-H.; Jensen, A. S.; Lohmann, M. S. & Nielsen, A. S. (2004), 'Comparative analysis of radio occultation processing approaches based on Fourier integral operators', *Radio Sci.* **39**(6), RS6004.

*Page 11, line 282: [Arnold1978] → (Arnold, 1978)*

Again rectified in LATEX. The LATEX citation command was inserted.

*Page 11, line 299 and page 12, line 304: [Gorbunov2019] is not listed in the reference section.*

It is, once again, a missing citation command. We carefully rechecked overall and found that further corrections of this type were needed for [Zou2019] and [Gorbunov2009a, Zou2019]. We hence also have corrected these.

*Page 19, line 407:*
*"COSMC-ECMWF" → "COSMIC-ECMWF"*

OK.

*Page 21, line 476 and 478: "Intoduction" → "Introduction"*
We found this to have been a typo in our BIBTEX data base, which also has been corrected.

**Anonymous Referee #2**

*"Here the difference metrics for β = 0 and optimal β cannot be directly compared, because they are evaluated over different statistical ensembles."*
*This makes the interpretation of the results presented in Figures 3-11 extremely difficult. The penetration depth alone does not seem to be a strong argument, particularly when β = 0 often provides more data above 1 km. It would be more useful to show the subset of refractivity values common to all retrievals.*

Following the suggestion of the Reviewer we performed some further study and found another important property of CT2A, as discussed below.

*In addition, the text says that the method mitigates systematic errors, but the metric shown in these figures combines systematic and random errors. I suggest that the systematic and random error estimates should be plotted separately. These points and the specific comments given below should be addressed before publication.*

The statement about the mitigation of the bias was made in a preliminary study, which was based on a much smaller volume of data. After the full study, we made a conclusion that it is the mean square difference between the RO and ECMWF refractivities that can be minimized by using the modified algorithm. Therefore, based on this additional finding, we refined the formulation in the abstract.

*Line 225: "don't" should be "do not".*

OK.

*Line 293: "The angular component of the momentum pd coincides with the ray impact parameter p, which is invariant in a spherically layered medium, but is perturbed by the horizontal gradients (Gorbunov and Lauritsen, 2009)". Healy (2001) also pointed this out.*

Healy (2001) refers to the technical report (Gorbunov, 1996), where the derivation of the impact parameter variation using the Hamiltonian form of ray trajectory equation was first presented.

*Line 299: [Gorbunov2019] not listed in references. Format of reference in text.*
*The references appear to change format e.g., line 306 "[Zou2019]" and line 310 "[Gorbunov2009a, Zou2019]". These should be (Zou et al., 2019) and (Gorbunov and Lauritsen, 2009).*

We corrected the references (cf. the similar remarks of Reviewer #1). That was related to technical corrections regarding the LATEX.

*Line 364: "co-located ECMWF refractivity profiles". It would be useful to give more detail here. For example, does this computation include the tangent point drift? Do you compute the refractivity directly from the ECMWF P, T and Q fields? Are they ECMWF forecasts or analyses? What resolution?*

We used ECMWF analyses at 1-degree latitudinal and 1-degree longitudinal resolution, with 91 vertical level covering the altitude range up to about 80 km. The refractivity was evaluated from pressure, temperature, and humidity fields. The

tangent point drift was taken into account. We checked that this is also noted in the manuscript so that it is clear to the readers.

*Line 366: It would be useful to split this metric into to systematic and random errors instead of combining them, particularly if the transform is likely to improve systematic errors, as noted in the abstract.*

We preferred to correct the statement about the systematic errors.

*Line 373: "The CT2A algorithm also improves the penetration increasing the number of data in the altitude range below 0.5 km."*
*This is correct, but $\beta = 0$ appears to provide more data above 1 km. Why is this? Are you using the transformed amplitude to cut-off the data? Please explain.*

This is linked to the QC procedure and still needs further investigation that will be performed beyond the scope of this initial introduction study of the CT2A.

*Line 374: "Here the difference metrics for $\beta = 0$ and optimal $\beta$ cannot be directly compared, because they are evaluated over different statistical ensembles.".*
*This really makes it difficult for the reader to judge whether the new transform is an advantage or not in all the subsequent figures. Is it possible to present the results for a dataset common to all $\beta$ values to help the reader interpret the results?*

As noted above, we evaluated the statistics for the common dataset and found another important property of CT2A. 
[revised manuscript text omitted]

135 The stationary phase point $t_s\left(p\right)$ of this integral satisfies the equation:

$$\frac{\partial}{\partial t} S_2\left(p,t\right) + \dot{\Psi}\left(t\right) = 0.\tag{5}$$

Accordingly, the transformed field, under the assumption that the Eq. (5) has a single solution $t_s\left(p\right)$, is also quasi-monochromatic and can be written as follows:

$$\hat{u}\left(p\right) = A^{'}\left(p\right) \exp\left(ik\Psi^{'}\left(p\right)\right) = A^{'}\left(p\right) \exp\left(ik\left(S_2\left(p,t_s\left(p\right)\right) + \Psi\left(t_s\left(p\right)\right)\right)\right).\tag{6}$$

140 Its  instantaneous frequency equals:

$$\xi\left(p\right) = \dot{\Psi}^{'}\left(p\right) = \frac{d}{dp}\left(S_2\left(p,t_s\left(p\right)\right) + \Psi\left(t_s\left(p\right)\right)\right) =$$
$$= \frac{\partial}{\partial p} S_2\left(p,t_s\left(p\right)\right) + \left(\frac{\partial}{\partial t} S_2\left(p,t_s\left(p\right)\right) + \frac{\partial}{\partial t}\Psi\left(t_s\left(p\right)\right)\
[revised manuscript text omitted]

---

## Author Response (AR2)

*1. $\beta \neq 0$ just seems to produce a different sample through stricter QC. The authors show that when the calculations with $\beta = 0$ are compared with $\beta \neq 0$ for the same sample, the statistical results are the same. I think the reader needs to understand Why the sample numbers are different for $\beta = 0$ and $\beta \neq 0$.*

We have analyzed our algorithm and found a mistake in the estimation of the determination of the shadow border for $\beta \neq 0$. After the correction of the mistake, the number of samples for different values of $\beta$ is no longer that different. Still, the first statement of the reviewer remains correct. CT2A does improve the statistics by using a stricter QC, as explained below.

*2. What specific QC criteria in the processing is leading to this difference? If $\beta \neq 0$ is actually better than $\beta = 0$, wouldn't this be expected to produce better statistics when computed for the same sample?*

The CT2A algorithm results in a better estimate of the shadow border, discriminating weaker pieces of the ray manifold.

*3. Do these results justify an operational processing change from $\beta = 0$ to $\beta \neq 0$?*

This question is beyond the scope of this paper. Implementing any changes in the operational processing, or in production is a hard decision, especially, when the improvement in accuracy is achieved by the reduction of the penetration depth, which is considered to be a very important characteristic of any data processing system. In any event, such a decision must inevitably be preceded by an impact study.